# Effects of Different Nitrogen Fertilizer Rates on Spring Maize Yield and Soil Nitrogen Balance Under Straw Returning Conditions of Cold Regions

**DOI:** 10.3390/plants14193087

**Published:** 2025-10-07

**Authors:** Jinghong Ji, Shuangquan Liu, Xiaoyu Hao, Yu Zheng, Yue Zhao, Yuqi Xia, Zhanqiang Xing, Wei Guo

**Affiliations:** 1Heilongjiang Academy of Black Soil Conservation and Utilization, Harbin 150086, China; shuangquanliu@126.com (S.L.); xiaoyuhao1981@sina.com (X.H.); annadian@163.com (Y.Z.); zhaoyue2108@163.com (Y.Z.); 2Key Laboratory of Black Soil Protection and Utilization, Ministry of Agriculture and Rural Areas, Harbin 150086, China; 3Institute of Life Sciences, Heilongjiang Bayi Agricultural University, Daqing 163319, China; 13845008701@163.com; 4Heilongjiang Academy of Agricultural Sciences, Harbin 150086, China; xingzhanqiang@163.com (Z.X.); guoweixinwei@126.com (W.G.)

**Keywords:** straw returning, nitrogen balance, nitrogen fertilizer, spring maize yield, black soil of cold regions

## Abstract

Under the condition of straw returning to the field, appropriate nitrogen fertilizer application is one of the key factors used to improve crop yield and ensure environmental safety. Therefore, an experiment with different rates of nitrogen fertilization was conducted with a randomized block design in Harbin, China. The straw was deeply plowed back into the field after harvest in the autumn. The nitrogen application rates were 0, 75, 150, 180, 225, and 300 kg·ha^−1^. The purpose of this study is to clarify the appropriate amount of nitrogen fertilizer under the condition of straw returning to the field and to provide technical support for high-yield and high-efficiency maize in cold regions. The results indicated that the yield of maize first increased and then stabilized as the amount of nitrogen fertilizer increased, while the economic benefits first increased and then decreased. When the nitrogen application rate exceeds 225 kg·ha^−1^ or is lower than 150 kg·ha^−1^, the economic benefits significantly decrease. When high-nitrogen fertilizer rates of 225 kg·ha^−1^ and 300 kg·ha^−1^ were applied, the residual nitrate nitrogen in the soil was increased by 2.1 times and 2.3 times, respectively, compared to before sowing. With the increase in the nitrogen application rate, the nitrogen fertilizer utilization efficiency and agronomic efficiency decreased, and the apparent nitrogen loss and nitrogen surplus significantly increased. Comprehensively considering the maize yield, benefits, and environmental risk factors the suitable nitrogen application rate was in a range of 170.2 kg·ha^−1^ to 178.2 kg·ha^−1^ in the first year and 150.0 kg·ha^−1^ to 171.3 kg·ha^−1^ in the second year. This work provides a theoretical basis and technical support for the rational application of nitrogen fertilizer and high-yield and high-efficiency spring maize under the condition of straw returning to the field.

## 1. Introduction

Straw is a valuable agricultural renewable resource. Crop straw contains abundant organic matter, nitrogen, phosphorus, potassium, and micro-elements. Returning straw to the field can increase soil fertility, improve soil structure, enhance nitrogen fertilizer effectiveness, and promote sustainable agricultural development [1,2]. Heilongjiang Province is the northernmost province in China, with lower temperatures in spring. The planting area of crops is large, with an annual straw production of about 90 million tons and a collection capacity of about 77 million tons, accounting for about 1/10 of the national total and more than 40% of three provinces and one region in Northeast China [3]. In recent years, with the implementation of black soil protection policies and the demand for national food security, the area of straw returning to the field has been continuously expanding. The straw returning rate in Heilongjiang Province is expected to reach nearly 70% by 2024.

Maize is one of the major grain crops in Heilongjiang Province. The sown area of maize reached 5.90 million hectares in 2024, accounting for 40% of the total area of grain crops in Heilongjiang Province and 13.19% of the maize planting area of China [4]. In general, the C/N of the straw of Gramineae crops is 60 to 80, and the optimum C/N for the growth of soil microorganisms is 20 to 30 [5]. Due to the large biomass of maize stalks and their high carbon-to-nitrogen ratio, after being returned to the field, the decomposition process caused by microorganisms consumes available nitrogen (mineral nitrogen) in the soil, leading to the short-term “biological immobilization” of soil nitrogen, which may cause nitrogen deficiency in crops. In the later stage of stalk decomposition (about 60 days later), the death of microorganisms releases the immobilized nitrogen, forming a “slow-release nitrogen source” [6]. The release of nitrogen from stalks requires a certain period of decomposition before it can be absorbed and utilized by crops [7]. Reducing the amount of nitrogen fertilizer directly can affect the normal growth of crops [8,9]. An appropriate amount of nitrogen fertilizer is important for high-yield and high-efficiency maize production, but when the amount of nitrogen fertilizer exceeds a certain threshold, it is not conducive to increasing maize yield and may also lead to wasted nitrogen fertilizer and potential environmental risks [10,11,12].

The decomposition of straw is affected by climatic and geographical environmental factors, among other things [13]. Li Changming et al. [14] studied the straw decomposition process of wheat and maize. The nitrogen and phosphorus in straw showed characteristics of first enrichment and then release in cold temperate zones, and the characteristics of direct release in warm temperate and subtropical zones. The degree of straw decomposition and nutrient release is closely related to the amount of nitrogen fertilizer applied in particular [15]. The use of nitrogen fertilizer needs to be adjusted according to the different release rates of straw nutrients. Currently, there are many studies on the effects of nitrogen fertilizer application under the condition of straw returning to the field on maize yield, nitrogen fertilizer utilization rates [16,17], and nitrogen balance [18,19,20,21], but these studies mainly focus on regions with relatively high temperatures and fertilizer application rates. There are few studies on the impact of the nitrogen nutrient balance on soil under the nitrogen fertilizer application of spring maize in cold regions. Therefore, an experiment was conducted in Harbin, the main maize production area in Heilongjiang Province, from 2023 to 2024. By studying the effects of nitrogen application under the condition of straw returning to the field on the nitrogen balance of black soil and the yield and benefits of maize, this research aims to provide a theoretical basis and technical support for the rational application of nitrogen fertilizer to spring maize and promote sustainable agricultural development.

## 2. Results

### 2.1. Maize Yield and Economic Benefits

The results showed that the yield of maize first increases and then stabilizes as the amount of nitrogen fertilizer increases, while the economic benefits first increase and then decrease (Table 1). Compared with the no-nitrogen fertilizer application treatment (N0), the yield increase rates of the N75, N150, N180, N225, and N300 treatments were 31.11%, 51.17%, 57.33%, 62.00%, and 56.62%, respectively, in 2023, which were 26.80%, 38.58%, 53.08%, 46.75%, and 48.58%, respectively, in 2024. The nitrogen application rates of 180 kg·ha^−1^ and 225 kg·ha^−1^ had significantly higher benefits than the other treatments in 2023; the nitrogen application rate of 180 kg·ha^−1^ had significantly higher benefits than the other treatments in 2024. This indicates that both excessive and insufficient nitrogen application pose risks of reduced yield and provide benefits to a certain extent. When the nitrogen application rate exceeded 225 kg·ha^−1^ or was lower than 150 kg·ha^−1^, the economic benefit significantly decreased (Table 1).

The relationship between maize yield and the nitrogen application rate (N) conforms to a linear plateau model (Figure 1). According to this equation, the maximum maize yield was 11,751 kg·ha^−1^, and the optimal nitrogen application rate was 178.2 kg·ha^−1^ in 2023; the maximum maize yield was 12,135 kg·ha^−1^, and the required nitrogen application rate was 182.2 kg·ha^−1^ in 2024 (Figure 1). Considering the experimental results for 2023 and 2024, it can be seen that this value is close to the nitrogen application rate of 180 kg·ha^−1^ designed in this experiment under the condition of straw returning to the field.

### 2.2. Nitrogen Fertilizer Utilization Rate and Agronomic Efficiency

All treatments were carried out under the condition of returning straw to the field; therefore, the nitrogen brought into the soil by straw returning was not included in the calculation of nitrogen fertilizer utilization efficiency. Only the nitrogen brought into the soil by fertilization was considered. The results show that the nitrogen fertilizer utilization efficiency and agronomic efficiency were decreased with the increase in the nitrogen application rate (Figure 2 and Figure 3). The range of nitrogen fertilizer utilization efficiency was 30.1%~52.4% in 2023, and the variation range of agronomic efficiency was 13.9 kg·kg^−1^ ~30.6 kg·kg^−1^. The range of nitrogen fertilizer utilization efficiency in 2024 was 27.1%~55.8%, and the variation range of agronomic efficiency was 13.2 kg·kg^−1^~29.0 kg·kg^−1^. Compared with treatment N75, the nitrogen fertilizer utilization efficiency was decreased by 7.5 and 14.8 percentage points (*p* < 0.05) under treatment N180 in 2023 and 2024. When the nitrogen application rate was 300 kg·ha^−1^, it was decreased by 22.3 and 28.7 percentage points (*p* < 0.05); the agronomic efficiency of nitrogen fertilizer significantly decreased by 11.24 g·kg^−1^ and 7.7 kg·kg^−1^ compared to treatment N150 in 2023 and 2024. Excessive nitrogen application has difficulty achieving high nitrogen fertilizer utilization efficiency, and selecting an appropriate nitrogen application rate is crucial for maintaining high maize yield and increasing nitrogen fertilizer utilization efficiency.

### 2.3. Changes in Inorganic Nitrogen in Soil Profiles

Under the condition of returning straw to the field, the accumulation of mineral nitrogen was investigated in different soil layers. The results indicated that the accumulation of mineral nitrogen in the soil showed an increasing trend at harvest with the increase in the nitrogen application rate (Figure 4). Compared with the soil at the beginning of the experiment in 2023, the residual mineral nitrogen in the N0 and N75 treatments was decreased by 71.5% and 26.6%, respectively, while under treatments N180, N225, and N300, it increased by 24.6%, 69.0%, and 96.3%, respectively. In particular, the residual mineral nitrogen was increased significantly under treatments N225 and N300. High fertilization not only increased the mineral nitrogen content in the surface soil but also enhanced the leaching of nitrogen into deeper soil layers. In the 80–100 cm soil layer, under treatments N225 and N300, the residual nitrate nitrogen in the soil was increased by 2.1 times and 2.3 times, respectively, compared to before sowing. In the 0–100 cm soil layer, the residual nitrate nitrogen in the N0, N75, N180, N225, and N300 treatments was 57.20 kg·ha^−1^, 94.89 kg·ha^−1^, 119.24 kg·ha^−1^, 137.00 kg·ha^−1^, 170.36 kg·ha^−1^, and 193.03 kg·ha^−1^, respectively (Figure 4).

### 2.4. Effects of Nitrogen Application Rates on Nitrogen Balance

The apparent nitrogen balance in the 0–100 cm soil layer is shown in Table 2. The results indicate that the apparent nitrogen loss and nitrogen surplus significantly increased with the increase in the nitrogen application rate. Under straw returning conditions, the apparent nitrogen losses under treatments N75, N150, N180, N225, and N300 were 2.3 kg·ha^−1^, 15.8 kg·ha^−1^, 15.3 kg·ha^−1^, 21.7 kg·ha^−1^, and 57.3 kg·ha^−1^, respectively in 2023 and 8.1 kg·ha^−1^, 41.4 kg·ha^−1^, 47.2 kg·ha^−1^, 58.9 kg·ha^−1^, and 138.6 kg·ha^−1^, respectively, in 2024. The nitrogen application rates of 225 kg·ha^−1^ and 300 kg·ha^−1^ did not significantly increase the nitrogen uptake by plants compared with the nitrogen application rate of 180 kg·ha^−1^, but the apparent nitrogen loss significantly increased (*p* < 0.05). This indicated that excessive nitrogen fertilizer application not only fails to promote crop nitrogen uptake but also leads to a sharp increase in nitrogen loss.

Among the nitrogen output items, both the residual mineral nitrogen in the soil and the apparent nitrogen loss significantly increased with the increased nitrogen application rate, and both were significantly and positively correlated with the nitrogen application (Figure 5 and Figure 6). A regression analysis was conducted on the residual mineral nitrogen in the soil and the nitrogen application rate in 2023; the regression equation was *y* = 0.5119*x* + 90.399 (*R*^2^ = 0.9906, *p* < 0.01), indicating that for every 1 kg increase in the nitrogen application rate, the residual mineral nitrogen in the soil increased by 0.5119 kg (Figure 5). A regression analysis was also conducted on the residual mineral nitrogen in the soil and the nitrogen application rate in 2024, and the regression equation was *y* = 0.5722*x* + 89.344 (*R*^2^ = 0.9830, *p* < 0.01), indicating that for every 1 kg increase in nitrogen application rate, the residual mineral nitrogen in the soil increased by 0.5722 kg (Figure 5). Nitrogen apparent loss was positively correlated with nitrogen application rate in a quadratic equation. From Figure 6, we can calculate that the amount of nitrogen apparent loss was 29.3 kg·ha^−1^ and 56.7 kg·ha^−1^ when applying 225 kg·ha^−1^ and 300 kg·ha^−1^ in 2023, respectively. The amount of nitrogen apparent loss was 73.7 kg·ha^−1^ and 133.7 kg·ha^−1^ when applying 225 kg·ha^−1^ and 300 kg·ha^−1^ in 2024, respectively.

### 2.5. The Relationship Between Nitrogen Surplus Rate, Nitrogen Application Rate, and Nitrogen Loss Amount

This study calculated the apparent nitrogen surplus rates for treatments N75, N150, N180, N225, and N300 based on the data in Table 2, which were −75.43%, −35.04%, −7.04%, 3.34%, 20.36%, and 55.12% in 2023, respectively; the apparent nitrogen surplus rates in 2024 were −64.65%, −20.78%, 12.99%, 23.66%, 43.58%, and 79.15%, respectively. It can be seen that the nitrogen surplus rate showed an increasing trend under the same nitrogen fertilizer application with the increase in straw-returning years. When the nitrogen fertilizer application was 180 kg·ha^−1^, the nitrogen surplus rate was only 3.34% in the first year of straw returning. The nitrogen surplus rate reached 23.66% in the second year of straw returning.

The nitrogen surplus rate was significantly correlated with nitrogen application rate, the nitrogen fertilizer utilization rate, and nitrogen loss. There was a highly significant linear positive correlation between the nitrogen surplus rate and the nitrogen application rate (Figure 7), a negative correlation with the nitrogen fertilizer utilization rate (Figure 8), and a highly significant quadratic curve relationship with nitrogen loss (Figure 9). According to the formula in the figure, it can be calculated that. When the nitrogen fertilizer input and crop nitrogen absorption were equal, the surplus rate was zero, the nitrogen application rate was 170.2 kg·ha^−1^, the nitrogen recovery rate was 44.8%, and the apparent nitrogen loss was 21.3 kg·ha^−1^ in 2023, and the nitrogen application rate was 128.8 kg·ha^−1^, the nitrogen recovery rate was 48.2%, and the apparent nitrogen loss was 21.7 kg·ha^−1^ in 2024.

## 3. Discussion

### 3.1. Environmental Effects of Residual Nitrate Nitrogen in Soil Profiles

The residual soil inorganic nitrogen in dryland was mainly in the form of nitrate nitrogen [22,23]. A suitable and balanced supply of nitrate nitrogen during the growth period is beneficial to the nutrient requirements of maize [24]. However, excessive nitrate nitrogen is prone to leaching with water, posing potential environmental risks [25,26,27]. Guo found that more than 50% of the nitrogen in China’s groundwater comes from the leaching of nitrogen from farmland [28]. Xiao’s research results showed that there is an exponential relationship between the amount of nitrate nitrogen leaching and the amount of nitrogen fertilizer applied, and the amount of nitrate nitrogen leaching will increase rapidly when the amount of nitrogen fertilizer applied exceeds this value [29]. Kong’s research results showed that the content of soil NO_3_^−^-N was significantly higher under treatments N280 and N350 than under N70, N140, N210, and other nitrogen application treatments in the 40–100 cm soil layer [30]. The results of this study also show that high fertilization not only increases the mineral nitrogen content in surface soil but also enhances the leaching of nitrogen to deeper soil layers. When the nitrogen application rates were 225 kg·ha^−1^ and 300 kg·ha^−1^, the residual amounts of soil nitrate nitrogen in the 80–100 cm soil layer increased significantly by 2.1 times and 2.3 times compared to before sowing. Suitable cumulative nitrate nitrogen content in the soil profile can be maintained by adjusting the application amount of nitrogen fertilizer. This research result is consistent with previous research results [31,32,33].

Jemison and Fox found that the accumulation of nitrate nitrogen in the soil after harvest is positively correlated with the nitrate nitrogen content in the soil solution. Keeping the residual nitrate nitrogen in the soil within a certain range after harvest is an important indicator for balancing yield and the environment [34]. Liu found that the residual value of inorganic nitrogen in the soil, taking into account the absorption and utilization by summer crops, should be 100 kg·ha^−1^ [35]. The European Union set the residual nitrate nitrogen in soil to 90 kg·ha^−1^ after autumn harvests [36]. This indicator was formulated from the perspective of environmental protection without pursuing high yields. It was not in line with the actual situation in China, where high yields are pursued. Zhong et al. [37] combined the actual situation of protecting the environment and achieving high yields and suggested that the residual nitrate nitrogen in the soil in the northern winter wheat–summer maize rotation area of China should not exceed 150 kg·ha^−1^ in the 0–90 cm soil layer. The residual nitrate nitrogen for high-yield and high-efficiency maize cultivation in Jilin Province should be less than 150 kg·ha^−1^ [38]. Through this experiment, it can be seen that when straw is returned to the field in the first two years, the nitrogen application rate exceeds 225 kg·ha^−1^, and the residual nitrate nitrogen in the soil is 170.36 kg·ha^−1^, which is greater than 150 kg·ha^−1^ and exceeds the environmental carrying capacity. As the number of consecutive years of spring maize cultivation increases, the accumulation of soil nitrate nitrogen will increase year by year. Therefore, continuous attention should be paid to the appropriate nitrogen application rate and the change in nitrate nitrogen content under long-term straw return conditions to provide technical support for achieving high-yield and high-efficiency maize production without causing excessive accumulation or loss of nitrate nitrogen in the soil layer and achieving a balance between economic and environmental benefits.

### 3.2. Appropriate Nitrogen Fertilizer Application Rate for Maize Under Straw Returning Conditions in Cold Black Soil Areas

Determining the appropriate nitrogen fertilizer application rate for crops requires balancing aspects such as yield, economic benefits, and the environment. In addition to considering the yield-increasing effect and nitrogen use efficiency, the level of residual inorganic nitrogen in the soil should also be taken into account [39]. Appropriate nitrogen fertilizer can improve the microenvironment of the soil and regulate the carbon–nitrogen ratio of the soil, thereby accelerating the decomposition of straw and release of nutrients, providing sufficient nutrient supply for corn growth and ultimately increasing corn yield [40]. Some studies have suggested that when the nitrogen balance surplus in farmland exceeds 20.0%, it may pose a potential threat to the environment [41,42,43]. In this study, through equation fitting and calculation, it was determined that when the surplus rate of nitrogen was zero, the nitrogen application rates were 170.2 kg·ha^−1^ and 128.8 kg·ha^−1^, respectively, in 2023 and 2024. If the nitrogen fertilizer application rates exceed 217.4 kg·ha^−1^ and 171.3 kg·ha^−1^ (Figure 7), the soil nitrogen surplus will be higher than 20%. We can also see that the nitrogen application rates for the highest maize yield were 178.2 kg·ha^−1^ and 182.2 kg·ha^−1^, respectively, in 2023 and 2024 (Figure 1). When the nitrogen application rate exceeds 225 kg·ha^−1^ or is lower than 150 kg·ha^−1^, the economic benefits are significantly decreased (Table 1). Therefore, considering the maize yield, benefits, soil nitrate nitrogen content, nitrogen fertilizer surplus, and environmental risk factors comprehensively, the nitrogen fertilizer application rates were more suitable within a range of 170.2~178.2 kg·ha^−1^ in the first year (in 2023) and 150.0~171.3 kg·ha^−1^ in the second year of returning maize straw to the field (in 2024). The maize yields were 11564~11751 kg·ha^−1^ and 11462~11907 kg·ha^−1^ for this amount of nitrogen fertilizer (Figure 1): the surplus rates of nitrogen were 0~3.39% and 9.97~20.0% (Figure 7); the nitrogen recovery rates were 44.8~43.9% and 48.2~42.6% (Figure 8); and the apparent nitrogen losses were 21.3~23.1 kg·ha^−1^ and 30.6~41.1 kg·ha^−1^ (Figure 9).

Due to different soil types, climatic characteristics, and soil environments, the suitable nitrogen application amount for straw returning to the field in different regions is also different. Kong established the relationship between the amount of nitrogen fertilizer and the yield of maize, the apparent nitrogen recovery rate, the residual amount of inorganic nitrogen in the soil, and the apparent amount of nitrogen loss, determining that the appropriate amount of nitrogen fertilizer for one-time application was 198–219 kg·ha^−1^. At this time, this can not only maintain a high yield of maize but also maintain the basic stability of the mineral nitrogen pool before and after maize harvest in the 0–100 cm soil layer, and the apparent amount of nitrogen loss is also maintained at a low level [26]. Zhang conducted research on spring maize planting areas with different soil fertility levels in Shanxi Province and concluded that the safe nitrogen application rates for different soil fertility levels vary, with 270 kg·ha^−1^ being suitable for medium- and low-fertility fields and 180 kg·ha^−1^ being suitable for medium- and high-fertility fields [44]. Chen believed that the appropriate nitrogen application rate for high-yield and high-efficiency maize cultivation in Jilin Province should be 195 kg·ha^−1^ to 225 kg·ha^−1^, and a nitrogen application rate of 195 kg·ha^−1^ had significant economic and environmental benefits [38]. Wang’s research results showed that the suitable amount of nitrogen fertilizer for maize was 179.5~198.4 kg·ha^−1^ under continuous straw returning to the field in central northeast black soil [20]. Compared with the above research, the suitable amount of nitrogen fertilizer in this study area was lower, mainly because the straw returning brought some of the nutrients; on the other hand, the tested black soil was relatively fertile. Thus, the amount of nitrogen fertilizer was relatively low.

This study also found that the nitrogen surplus rate shows an increasing trend with the increase in the years of straw returning under the same nitrogen fertilizer application rate. When the nitrogen fertilizer application rate was 180 kg·ha^−1^, the nitrogen surplus rate was only 3.34% in the first year of straw returning, while in the second year, the nitrogen fertilizer surplus rate reached 23.66%. This was mainly due to the higher total nitrogen input in 2024 (427.9 kg/ha) compared to that in 2023 (466.6 kg·ha^−1^). Although the nitrogen fertilizer application rate was the same at 180 kg·ha^−1^, the nitrogen content provided in 2024 was higher than that in 2023, resulting in an increase in nitrogen surplus in 2024 (234.0 kg·ha^−1^ in 2024 and 194.5 kg·ha^−1^ in 2023). This also indicated that if maize straw is returned to the field for a long period, the nitrogen fertilizer application rate can be gradually reduced. The research results are consistent with those of previous studies [7,20,45]. The reduction range for each year still needs to be established through long-term and stable positioning experiments and monitoring, which is also the work we will continue to carry out in the future. 

## 4. Materials and Methods

### 4.1. Study Site

This experiment was conducted from 2023 to 2024, and the experimental area is located in the Minzhu Science and Technology Park of the Heilongjiang Academy of Agricultural Sciences (45°51′ N, 126°50′ E) of Harbin City, Heilongjiang Province, China. This area has a cold temperate continental monsoon climate; elevation is 127 m, the annual average temperature is 3.5 °C, the annual average precipitation is 545.7 mm, and the annual average frost-free period is 135 days. The soil is typical black soil. The soil fertility is medium to high. The contents of total organic matter, alkaline nitrogen, available phosphorus, and available potassium were 28.4 g·kg^−1^, 129.8 mg·kg^−1^, 12.3 mg·kg^−1^, and 202.5 mg·kg^−1^, respectively, and the pH was 7.10. The soil bulk densities of the soil layers at 0–20 cm, 20–40 cm, 40–60 cm, 60–80 cm, and 80–100 cm are 1.21 g·cm^−3^, 1.25 g·cm^−3^, 1.34 g·cm^−3^, 1.31 g·cm^−3^, and 1.32 g·cm^−3^, respectively. The nitrate nitrogen contents were 23.39 mg·kg^−1^, 11.37 mg·kg^−1^, 7.48 mg·kg^−1^, 5.15 mg·kg^−1^, and 7.29 mg·kg^−1^, respectively; the ammonium nitrogen contents were 25.01 mg·kg^−1^, 29.22 mg·kg^−1^, 23.34 mg·kg^−1^, 18.73 mg·kg^−1^, and 11.28 mg·kg^−1^, respectively.

### 4.2. Experimental Design

Trials with different rates of nitrogen fertilizer application were set, and a randomized block design was used in 2023 and 2024. The nitrogen application rates (counted by N) for each treatment were 0, 75, 150, 180, 225, and 300 kg·ha^−1^. The plot area was 52 m^2^, with three replications for each treatment. Urea (46% N) was used as nitrogen fertilizer, superphosphate (46% P_2_O_5_) as phosphorus fertilizer, and potassium chloride (60% K_2_O) as potassium fertilizer. In total, 50% of the nitrogen and potassium fertilizers were applied as base fertilizers and 50% as top dressings. All phosphorus fertilizers were applied as base fertilizers in one application. The application rates of the phosphorus and potassium fertilizers were 75 kg·ha^−1^ of P_2_O_5_ and 75 kg·ha^−1^ of K_2_O, respectively. The maize variety was Huanong 887 in 2023, with base fertilizer and sowing on 7 May, and nitrogen fertilizer was applied at the jointing stage on 28 June, with harvest on 24 September; the maize variety in 2024 was Kunrui 58, with base fertilizer and sowing on 7 May, and nitrogen fertilizer was applied at the jointing stage on 26 June, with harvest on 30 September. The length of two plats was 25.6 cm, and the row spacing was 65 cm. The amount of rainfall and the average monthly temperature are shown in Figure 10 and Figure 11.

The previous crop before the start of the experiment was corn, variety Huanong 887, and the amount of straw returned to the field was 10 t. The crop residues were deeply plowed back into the soil at a depth of 25–30 cm in the autumn of 2022 and finely mixed into the soil at a depth of 15–20 cm in the autumn of 2023, with the length of the residues being less than 10 cm. Based on the straw returning to the field in the previous year (2022), the measured nitrogen content of the straw, and the nitrogen release rate of the straw decomposition adopted from the research results of Gong Zhen [46], the nitrogen accumulation release rate in the first year of straw returning to the field was 52.5%, and the cumulative release rate in the second year of returning to the field was 70.66%. The main reason is that our experimental site and their experimental site were both in Harbin City, Heilongjiang Province, with basically the same latitude and longitude, climate, soil, etc.; many researchers have only studied the decomposition law of straw returning to the field in the year, but Gong Zhen conducted multi-year decomposition research on corn straw, which is suitable for this experiment for two consecutive years. Nitrogen from maize straw in 2023 and 2024 is shown in Table 3.

### 4.3. Soil and Plant Sampling Analysis, Crop Yield Measurement

To determine the nitrate nitrogen and ammonium nitrogen contents in the soil, soil samples were collected from 0–20 cm, 20–40 cm, 40–60 cm, 60–80 cm, and 80–100 cm layers using the S-point sampling method after the harvests in spring 2023 and autumn 2024. Soil samples were collected from the 0–20 cm layer in spring 2023 to determine the basic chemical properties. The soil was sampled from the ridge slope between two maize plants. Five points were mixed in each plot; the soil samples were immediately air-dried after being brought back and then sieved according to the requirements of the determination items, fully mixed, and preserved. The conventional method was used to analyze the basic physical and chemical properties of the soil [47]: soil nitrate nitrogen and ammonia nitrogen were extracted with 1 mol/L KCL and determined with an AA3 flow analyzer; soil organic matter was determined by the Qulin method; soil alkaline nitrogen was determined by the alkaline diffusion method; available phosphorus was determined by the Olsen method; available potassium was determined by the flame phot method; pH value was determined by the potentiometric method (soil-to-water ratio, 1:2.5); and soil bulk density was determined by the ring knife method.

Five representative maize plants from each plot were collected after the maize matured. After the maize had matured, the middle areas of 5 representative plants in the plots were randomly removed from the marginal effect, and we decomposed the grain and straw. The plant and grain samples were dried at 105 °C for 30 min and then dried at 70 °C. After weighing, the total nitrogen content in the samples was determined by the Kjeldahl method. The nitrogen content of the plant is used to calculate the utilization rate of nitrogen fertilizer. Ears of 10 m^2^ area from the middle 2 rows of each plot were taken for yield measurement, threshing, and weighing, and the actual grain yields were calculated based on a moisture content of 14%.

### 4.4. Statistical Analysis and Calculated Formulation

#### 4.4.1. Statistical Analysis

Statistics and analysis were determined and conducted using the software Excel 2016 and SPSS 19.0; the LSD method was used to conduct variance analysis to test the significance of differences between different treatments.

#### 4.4.2. Calculate Formulation

Nutrient balance calculation adopts the apparent balance method, which is expressed by the difference between nutrient input and output. Precipitation, irrigation, and other nutrients brought in were not considered. Nutrient output includes the accumulation of nutrients in maize stalks and grains after maturity. The calculation formulas for fertilizer nitrogen utilization rate, apparent nitrogen loss, etc., are as follows [20,48]:(1)Apparent nitrogen surplus rate (kg·ha^−1^) = (the amount of applied nitrogen − nitrogen amount uptake by aboveground plant parts)/nitrogen amount uptake by aboveground plant parts × 100%. (Note: The amount of applied nitrogen here includes both fertilizer nitrogen and nitrogen from straw.)(2)Soil residual mineral nitrogen content (calculated in 20 cm layers) (kg·ha^−1^) = soil bulk density × 20 × mineral nitrogen concentration/10.(3)Apparent mineralization of nitrogen (kg ha^−1^) = nitrogen amount uptake by aboveground plant parts in no-nitrogen treatment + residual soil mineral nitrogen in no-nitrogen treatment − initial soil mineral nitrogen in no-nitrogen treatment.(4)Apparent nitrogen loss (kg·ha^−1^) = the amount of applied nitrogen + initial soil mineral nitrogen + apparent mineralization of nitrogen − nitrogen amount uptake by aboveground plant parts − accumulated soil mineral nitrogen at harvest.(5)Nitrogen fertilizer utilization rate (%) = (nitrogen uptake by crops in nitrogen application treatment − nitrogen uptake by crops in no-nitrogen treatment)/nitrogen fertilizer input × 100.(6)Nitrogen agronomy efficiency(kg·kg^−1^) = (yield in nitrogen application treatment − yield in no-nitrogen treatment)/nitrogen fertilizer input.(7)Economic benefits = maize yield × maize prices − fertilizer prices × the amount.

## 5. Conclusions

High nitrogen application rates reduced the yield benefits of maize and apparent nitrogen use efficiency and agronomic efficiency and increased the soil mineral nitrogen content, nitrogen surplus, and apparent nitrogen loss under the condition of returning straw to the field. The relationship between maize yield and the nitrogen application rate conformed to the linear plus plateau model; the nitrogen surplus rates and losses were significantly and linearly positively correlated with the nitrogen application rate, negatively correlated with the nitrogen fertilizer utilization rate, and significantly quadratically correlated with nitrogen loss. Comprehensively considering maize yield; economic benefits; and environmental risk factors, such as soil nitrate nitrogen content and nitrogen surplus, the suitable nitrogen application rate range for maize production in the medium- and high-fertility black soil area was 170.2~178.2 kg·ha^−1^ in the first year and 150.0~171.3 kg·ha^−1^ in the second year under the condition of returning straw to the field. Within this nitrogen application range, a higher maize yield and economic benefits can be achieved; the nitrate nitrogen content and nitrogen surplus are relatively reasonable, the apparent nitrogen loss is relatively low, and the soil nitrogen apparent balance can be maintained. This research can provide a data basis for local governments to coordinate straw returning to the field and formulate land management policies and can provide technical support for the implementation of key technologies for straw returning to the field. It is conducive to the promotion and implementation of straw returning policies.

## Figures and Tables

**Figure 1 plants-14-03087-f001:**
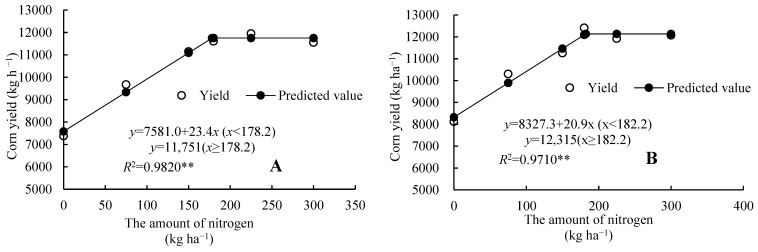
The relationship between nitrogen amount and maize yield. Note: (**A**,**B**) represent 2023 and 2024, respectively; the following is the same. “**” indicate highly significant differences at the 0.01 level under different treatments (*p* < 0.01).The same as follows.

**Figure 2 plants-14-03087-f002:**
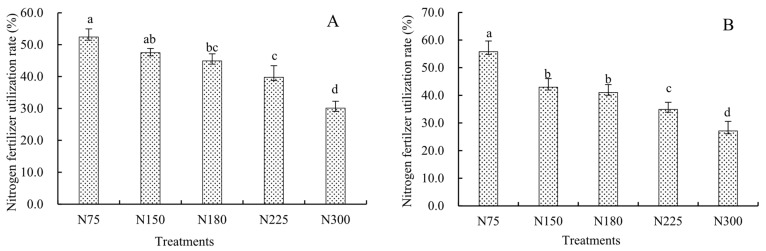
Nitrogen fertilizer utilization rate of maize under different treatments. Note: Different lowercase letters indicate significant differences at the 0.05 level under different treatments (*p* < 0.05). The same as follows.

**Figure 3 plants-14-03087-f003:**
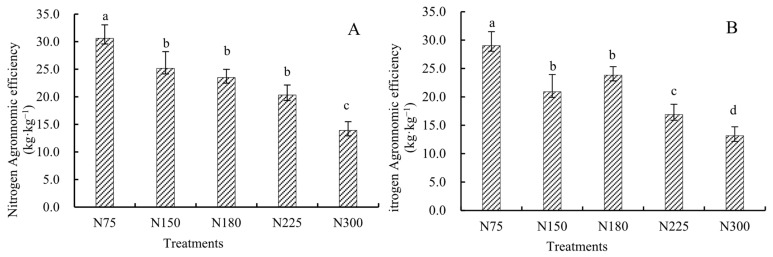
Nitrogen fertilizer agronomic efficiency of maize under different treatments.

**Figure 4 plants-14-03087-f004:**
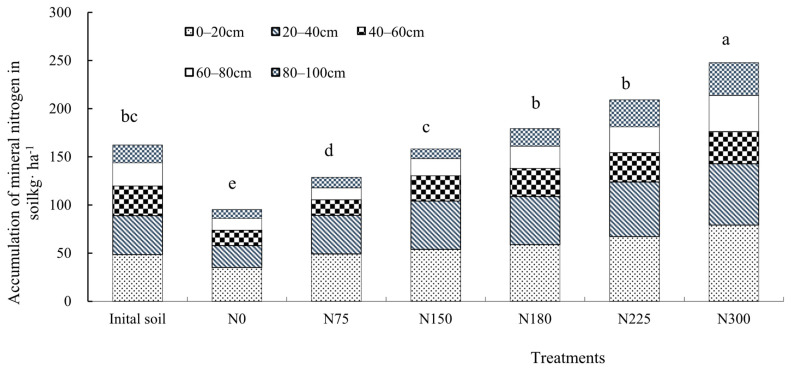
The influence of different nitrogen application rates on the accumulation of mineral nitrogen in the 0–100 cm soil profile.

**Figure 5 plants-14-03087-f005:**
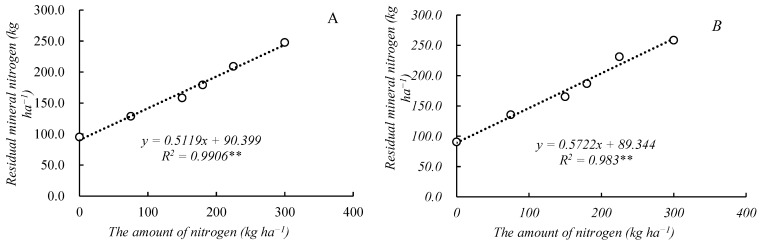
The relationship between nitrogen rate and residual mineral nitrogen.

**Figure 6 plants-14-03087-f006:**
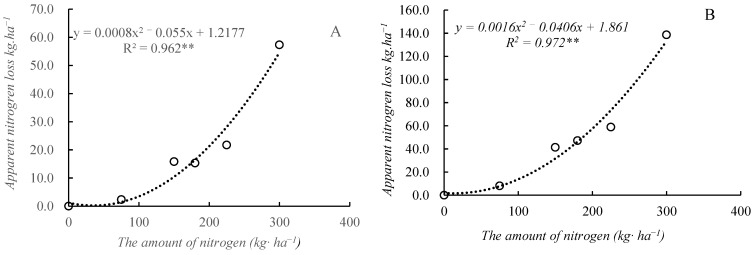
The relationship between nitrogen rate and apparent nitrogen loss.

**Figure 7 plants-14-03087-f007:**
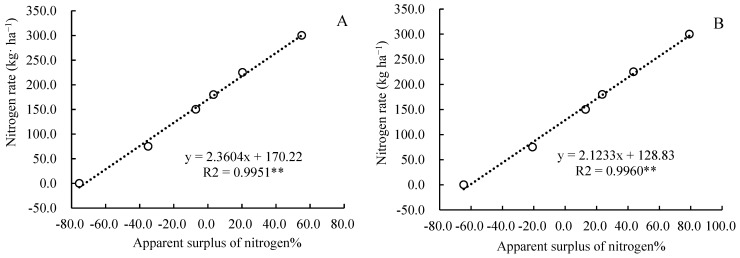
The relationship between nitrogen rate and the apparent surplus of nitrogen.

**Figure 8 plants-14-03087-f008:**
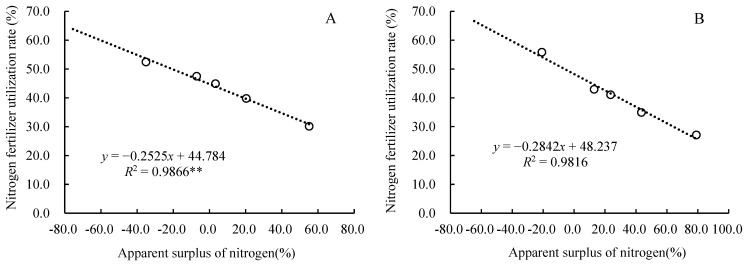
The relationship between nitrogen fertilizer utilization rate and the apparent surplus of nitrogen.

**Figure 9 plants-14-03087-f009:**
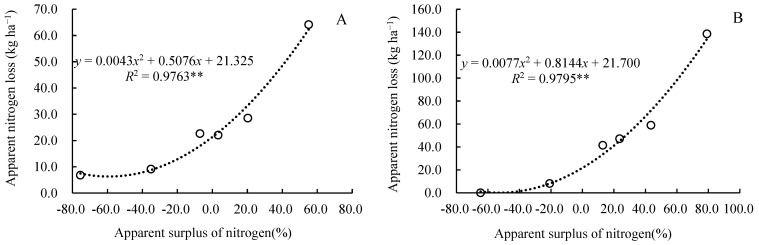
The relationship between apparent nitrogen loss and the apparent surplus of nitrogen.

**Figure 10 plants-14-03087-f010:**
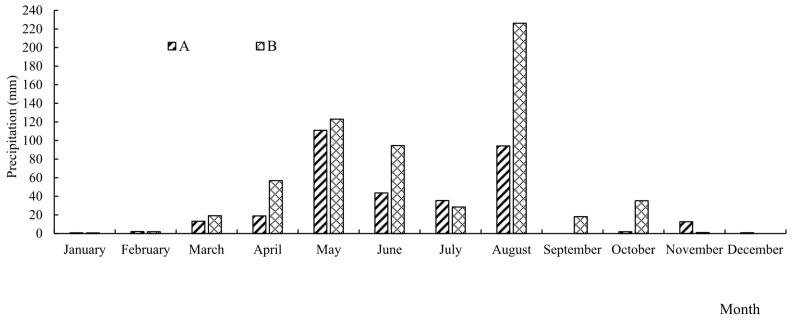
The precipitation of every month in the experimental region in 2023 and 2024.

**Figure 11 plants-14-03087-f011:**
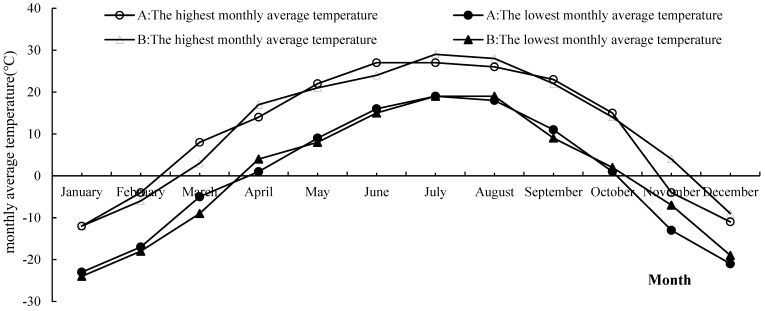
The highest and lowest monthly average temperatures in the experimental region in 2023 and 2024.

**Table 1 plants-14-03087-t001:** Maize yield and benefits under straw returning and nitrogen application.

Year	Treatment	Yield/ (kg·ha^−1^)	Increase Production/ (kg·ha^−1^)	Increase Rate/ %	Increase Economic Benefits/ (CNY·ha^−1^)
2023	N0	7376 d	—	—	—
	N75	9671 c	2295 c	31.11 c	4101 c
	N150	11150 b	3774 b	51.17 b	6570 b
	N180	11605 ab	4229 ab	57.33 ab	7284 a
	N225	11949 a	4573 a	62.00 a	7679 a
	N300	11552 ab	4176 ab	56.62 ab	6395 b
2024	N0	8124 d	—	—	—
	N75	10301 c	2177.2 c	26.80 c	2065 d
	N150	11259 b	3134.7 b	38.58 b	3074 c
	N180	12436 a	4287.9 a	53.08 a	4618 a
	N225	11922 a	3798.1 a	46.75 a	3648 b
	N300	12071 a	3946.9 a	48.58 a	3461 b

Note: The maize prices are CNY 2.00·kg^−1^, 1.48 ¥·kg^−1^ in 2023 and 2024; the prices of urea, heavy superphosphates, and potassium chloride are CNY 2.5·kg^−1^, CNY 4.3·kg^−1^, and CNY 3.5·kg^−1^, respectively. Artificial topdressing is CNY 750·ha^−1^. Different letters in the same column of the table represent the 5% significance level.

**Table 2 plants-14-03087-t002:** Soil nitrogen balance in the 0–100 cm layer during the growth period of maize (kg·ha^−1^).

Year	Items		N0	N75	N150	N180	N225	N300
2023	Input	(I) Nitrogen rate	0	75	150	180	225	300
		(II) Initial mineral nitrogen	162.2	162.2	162.2	162.2	162.2	162.2
		(III) Net nitrogen mineralization	55.6	55.6	55.6	55.6	55.6	55.6
		(IV) Nitrogen release from straw	30.1	30.1	30.1	30.1	30.1	30.1
		Total input (I + II + III + IV)	247.9	322.9	397.9	427.9	472.9	547.9
	Output	(V) Nitrogen uptake by maize	122.5 c	161.8 b	193.7 a	203.3 a	211.9 a	212.8 a
		(VI) Residual mineral nitrogen	95.3 e	128.7 d	158.3 c	179.3 c	209.2 b	247.7 a
		(VII) Apparent nitrogen loss	0	2.3 d	15.8 c	15.3 c	21.7 b	57.3 a
		Nitrogen surplus (VI + VII)	95.3 e	131.0 d	174.1 c	194.5 c	230.9 b	305.0 a
2024	Input	(I) Nitrogen rate	0	75	150	180	225	300
		(II) Initial mineral nitrogen	132.5	152.3	162.8	169.6	185.5	220.2
		(III) Net nitrogen mineralization	68.9	68.9	68.9	68.9	68.9	68.9
		(IV) Nitrogen release from straw	39.1	45.8	47.8	48.2	46.6	43.9
		Total input (I + II + III + IV)	240.5	342.0	429.4	466.6	526.0	633.0
	Output	(V) Nitrogen uptake by maize	110.6 c	152.5 b	175.0 ab	184.5 a	189.2 a	191.9 a
		(VI) Residual mineral nitrogen	90.8 d	135.6 c	165.2 bc	186.8 b	231.3 a	258.6 a
		(VII) Apparent nitrogen loss	0	8.1 d	41.4 c	47.2 bc	58.9 b	138.6 a
		Nitrogen surplus (VI + VII)	90.8 e	143.7 d	206.6 c	234.0 c	290.2 b	397.2 a

Note: Different letters indicate significant differences at the 0.05 level under different treatments (*p* < 0.05).

**Table 3 plants-14-03087-t003:** Nitrogen from maize straw in 2023 and 2024.

Treatments	2023			2024				
	Amount of Straw Input (kg·ha^−1^)	Nitrogen Content of Straw (%)	Nitrogen from Straw (kg·ha^−1^)	Amount of Straw Input (kg·ha^−1^)	Nitrogen Content of Straw (%)	Nitrogen from Straw in 2023 (kg·ha^−1^)	Nitrogen from Straw in 2022 (kg·ha^−1^) (II)	Nitrogen from Straw (kg·ha^−1^) (I + II)
0	10,000	0.571	30.1	9300	0.589	28.8	10.3	39.1
N75	10,000	0.571	30.1	10,762	0.627	35.5	10.3	45.8
N150	10,000	0.571	30.1	11,200	0.636	37.5	10.3	47.8
N180	10,000	0.571	30.1	11,750	0.612	37.9	10.3	48.2
N225	10,000	0.571	30.1	11,800	0.585	36.3	10.3	46.6
N300	10,000	0.571	30.1	11,700	0.545	33.6	10.3	43.9

Note: (1) The nitrogen from straw in 2023 = straw input amount × nitrogen content of straw input% × 52.65%. (2) the nitrogen from straw in 2024 includes two parts; one part is the nitrogen from straw in the same season (I = straw input amount × nitrogen content straw input% × 52.65%); the other part is the nitrogen from straw in the previous season (II = 10,000 × 0571% × (70.66% − 52.65%) = 10.3 kg/ha). Among them, 5265% is the first-year release rate of straw nitrogen, and 70.66% is the total release rate of straw nitrogen in the first and second years.

## Data Availability

The original contributions presented in this study are included in the article; further inquiries can be directed to the corresponding author.

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
