# Peer review of "Effects of Different Nitrogen Fertilizer Rates on Spring Maize Yield and Soil Nitrogen Balance Under Straw Returning Conditions of Cold Regions"

_plants, 2025, doi:10.3390/plants14193087_

Round 1

Reviewer 1 Report

Comments and Suggestions for Authors

While this is a nice manuscript, the manuscript needs further refinement. In particular english writing should be revised as there are several expressions, words and sentences that need review.

Materials and methods must also be improved. All the details of crop establishment need description, for example: hybrid name, panting dates, planting density, row distance, harvest date, and dates of specific relevant phenological stages.

There is little description of previous crop, straw amount and straw characteristics, factors that will be crucial to interpret results. Climate characteristics, accumulated rainfall and temperatures are also missing.

A major possible problem is related to harvest criteria. The authors mention that 60 ears were harvested (i.e. a fixed number). Given that the number of ears per plant likely vary with N rate (Parco et al., 2022) it is possible that harvest area varied among plots, thus making yield calculations wrong.

Given all the major concerns mentioned, I suggest reconsidering after making the suggested changes.

Comments on the Quality of English Language

The manuscript needs extensive review of english. In several parts of the text there are either expressions or words that are not appropriate, for example: "the reasonable application", "ecology environment", "hectar", "Six different nitrogen application gradients were", "blanched", and so on. 

Author Response

Dear Reviewer,

Thank you very much for your valuable comments on the paper. We have made modifications to the paper based on your comments, and the parts of the article that have been modified are marked in red font.Please review them. Thanks again!

 Description of modification are as follows

1.Open Review

1.1 Question: Materials and methods must also be improved. All the details of crop establishment need description, for example: hybrid name, panting dates, planting density, row distance, harvest date, and dates of specific relevant phenological stages.There is little description of previous crop, straw amount and straw characteristics, factors that will be crucial to interpret results. Climate characteristics, accumulated rainfall and temperatures are also missing.

Answer: Thank you for ponting this out. We agree with this comment. Materials and methods have improved in the article. (Modification was noted by red in 2.2. Experimental Design). Nitrogen from maize straw in2023 and 2024 were shown in table 1. Climate characteristics, accumulated rainfall and temperatures are shown in figure1 and figure 2.

1.2 Question: A major possible problem is related to harvest criteria. The authors mention that 60 ears were harvested (i.e. a fixed number). Given that the number of ears per plant likely vary with N rate (Parco et al., 2022) it is possible that harvest area varied among plots, thus making yield calculations wrong. Given all the major concerns mentioned, I suggest reconsidering after making the suggested changes.

Answer: The sampling area was 10 m2 in this experiment. Therefore“60 ears from the middle 2 rows of each plot were taken for yield measurement ”has been modifed by “ears of 10 m2 area from the middle 2 rows of each plot were taken for yield measurement”(lines 182-183).

1.3 Question: In particular english writing should be revised as there are several expressions, words and sentences that need review.Comments on the Quality of English Language. The manuscript needs extensive review of english. In several parts of the text there are either expressions or words that are not appropriate, for example: "the reasonable application", "ecology environment", "hectar", "Six different nitrogen application gradients were", "blanched", and so on.  

Answer: Modifications have been made to English writing. For example: "Appropriate nitrogen fertilizer" , "ological environment" , "hectare"(lines 65), "six different nitrogen application gradients","balanced",  etc.

Reviewer 2 Report

Comments and Suggestions for Authors

Manuscript ID: plants-3836351

Title: Effects of Different Nitrogen Fertilizer Rates on Spring Maize Yield and Soil Nitrogen Balance under Straw Returning Condition of Cold Regions

I have reviewed the manuscript titled “Effects of Different Nitrogen Fertilizer Rates on Spring Maize Yield and Soil Nitrogen Balance under Straw Returning Condition of Cold Regions” My evaluation is as follows:   

General comments

Authors carried out an experiment to determine to test the effect of nitrogen fertilization and maize straw in a cold climate. Straw was deep ploughed into the soil after harvest and then they applied six different nitrogen treatments. They found that maize yield first increased and then stabilized with amount of nitrogen fertilizer increases. Thus, applying high nitrogen fertilizer increases nitrogen content in surface soil and enhanced leaching thereby reducing the nutrient utilization efficiency. This study provides a theoretical basis for rational application of nitrogen fertilizer under the conditions of straw return to the soil. The study focus is topical, relevant and timely. It also fits well into Plant journal’s scope as it discusses an important issue in soil nutrient management and crop growth. The results are beneficial in utilization of nitrogen fertilizaers and can wrk towards nutrient use effciency. Authors need to clearly state the study objectives and provide an elaborate methodology and clarify on a few other issues before the paper can be considered for publication.

Specific comments

Title

The title is appropriate for the study subject

Abstract:

  • Authors have provided a good background information in the abstract. However, they need to provide the study objectives, and an elaborate methodology in brief. They have provided comprehensive results. They should only report only significant results and then provide the a sentence or two on the study implications and another concluding sentence.

Key words

  • They are sufficient.

Introduction

  • Authors have introduced the study by highlighting the value of straw as an agricultural resource by the roles it can play in supplying nutrients to the soil.
  • They then indicate how its abundance in the northern most part of China which is characterised by low temperatures during spring and maize cultivation.
  • These results in significant amounts of maize straw – a natural source of straw which can be returned back to the soil.
  • Authors have touched briefly on the carbon-nitrogen ratios in line 64-68. Since this affects decomposition, they should briefly elaborate these process to support the subsequent statements.
  • This should also be included in the methodology section – how did authors determine the amount of nutrients arising from straw deposition? This should be included in the methodology section.
  • Authors should also link this specifically to maize straw - as a source of nutrients that is mostly not accounted for in nutrient management plans and fertilization events. This will strengthen the study and help in the formulation of study objectives.
  • Authors indicate that there are many studies on the effect of nitrogen fertilizer under the conditions of straw returning to soils on maize yield, nitrogen fertilizer utilization rate. They should go beyond just mentioning. They should bring in the studies on temperature effects even though the studies were carried out in high temperature environments. This will bring out the importance of temperature and justify the study objecives.
  • The study objectives have not been clearly stated. Authors should phrase the study objectives to enhance clarity and measurability.

Materials and methods

  • The study location is well described in detail
  • Authors should elaborate on the experimental design structure. It is not clear how treatments were allocated to experimental units. Was it a completely randomized design, randomized complete block design or a split plot design? They should then provide statistical analysis on how these was analyzed.
  • It is also not clear how maize straw was incorporated into the experiment design. Was there only one treatment for maize straw?
  • Authors should indicate how soil sampling spots were identified. What about crop samples?
  • Formulars used in the calculation should be presented clearly.
  • They should also improve on the analysis. What was analyzed and how?

Results

  • Results are reported in detail.
  • Tables and figures are well presented and captioned.
  • What do authors mean by benefits in line 160 and several other instances and how were these benefits determined?
  • Authors indicate in line 192 that treatments were carried out under conditions of returning straw. This should be elaborated in the methodology section.
  • They should also provide the procedures for measuring nitrogen fertilizer utilization rates and agronomic efficiency.
  • Authors have provided results for several nitrogen loses (lines 244-246). Were these determined from this study? If yes, the procedures should be included in the methodology. If not, it should not appear in the results section. It will be beneficial to take this to the discussion section to enhance it by providing possible explanations of the study observations. These should also apply to several results reported here but may have not directly arised from this study such as nitrogen input fractions (line 247-249).

Discussion

  • Authors have discussed their results well. However, they have not discussed their implications and how they relate with other studies. They should not just report the results and cite references, but should include possible causes of the observed results, their implications and relationships with the cited references.

Conclusions, policy implications and recommendations.

  • Conclusions are supported by data.
  • Recommendation is provided.
  • Authors should give a statement on policy implications and future prospects of their study results.

References

  • The references are current and relevant. However, they are not sufficient and I suggest they bring more reference to help them relate their findings with other studies in the discussion section. They should also bring relevant references on my suggestion in the introduction.

Author Response

Dear Reviewer,

Thank you very much for your valuable comments on the paper. We have made modifications to the paper based on your comments, and the parts of the article that have been modified are marked in red font.Please review them. Thanks again!

 Description of modification are as follows

2.Open Review

2.1 Abstract: Authors have provided a good background information in the abstract. However, they need to provide the study objectives, and an elaborate methodology in brief. They have provided comprehensive results. They should only report only significant results and then provide the a sentence or two on the study implications and another concluding sentence.

Answer: Thank you for ponting this out. We agree with this comment. Abstract has been modified, which has been supplemented with the research objectives, research methods, simplified some conclusions, and summarized the of the study. ( Modification was noted by red in lines 12-48).

2.2 Introduction:Authors have touched briefly on the carbon-nitrogen ratios in line 64-68. Since this affects decomposition, they should briefly elaborate these process to support the subsequent statements. This should also be included in the methodology section – how did authors determine the amount of nutrients arising from straw deposition? This should be included in the methodology section. Authors should also link this specifically to maize straw - as a source of nutrients that is mostly not accounted for in nutrient management plans and fertilization events. This will strengthen the study and help in the formulation of study objectives. Authors indicate that there are many studies on the effect of nitrogen fertilizer under the conditions of straw returning to soils on maize yield, nitrogen fertilizer utilization rate. They should go beyond just mentioning. They should bring in the studies on temperature effects even though the studies were carried out in high temperature environments. This will bring out the importance of temperature and justify the study objecives. The study objectives have not been clearly stated. Authors should phrase the study objectives to enhance clarity and measurability.

Answer: Thank you for ponting this out. We agree with these comments. The introduction section has been modified and highlighted (lines 67-68,lines 81-88), and the nutrients produced after straw returning to the field are determinedin the Materials and Methods (Table 1). The study objectives have been clearly stated (lines 97-99)

2.3 Materials and methods: Authors should elaborate on the experimental design structure. It is not clear how treatments were allocated to experimental units. Was it a completely randomized design, randomized complete block design or a split plot design? They should then provide statistical analysis on how these was analyzed. It is also not clear how maize straw was incorporated into the experiment design. Was there only one treatment for maize straw? Authors should indicate how soil sampling spots were identified. What about crop samples? Formulars used in the calculation should be presented clearly. They should also improve on the analysis. What was analyzed and how?

Answer: Thank you for ponting this out. We agree with these comments. 1) This experiment was a randomized complete block design.2)Maize straw was incorporated into the experiment design, which was shown in Table1. 3)The collection method of soil and crop samplingspots, statistical analysis method has been supplemented in 2.2. Experimental Design, 2.3. Soil and Plant Sampling Analysis , Crop Yield Measurement,2.4 Statistical Analysis and Calculate Formulation.(lines 116-215)

.

2.3 Results: Results are reported in detail.Tables and figures are well presented and captioned.What do authors mean by benefits in line 160 and several other instances and how were these benefits determined?Authors indicate in line 192 that treatments were carried out under conditions of returning straw. This should be elaborated in the methodology section.They should also provide the procedures for measuring nitrogen fertilizer utilization rates and agronomic fficiency.Authors have provided results for several nitrogen loses (lines 244-246). Were these determined from this study? If yes, the procedures should be included in the methodology. If not, it should not appear in the results section. It will be beneficial to take this to the discussion section to enhance it by providing possible explanations of the study observations. These should also apply to several results reported here but may have not directly arised from this study such as nitrogen input fractions (lines 247-249).

Answer: 1)The "benefits" mentioned in line 160 and elsewhere are economic benefits, and the formula for supplementary calculation. Calculate Formulation of ecnomic benefit has been added. Ecnomic benefit=maize yield × the maize prices - fertilzer prices × the amount (lines 215). 2)The treatment was carried out on the condition of returning straw, which has been detailed in the method section(Table1). 3)Those result were not presented in this study has been deleted.

2.4 Discussion:Authors have discussed their results well. However, they have not discussed their implications and how they relate with other studies. They should not just report the results and cite references, but should include possible causes of the observed results, their implications and relationships with the cited references.

Answer: Some of discussion has been modified.(lines 424-481)

2.5 Conclusions, policy implications and recommendations: Conclusions are supported by data.Recommendation is provided.Authors should give a statement on policy implications and future prospects of their study results.

Answer:Related references have been added in Conclusions.(lines 499-503)

2.6 References:The references are current and relevant. However, they are not sufficient and I suggest they bring more reference to help them relate their findings with other studies in the discussion section. They should also bring relevant references on my suggestion in the introduction.

Answer:Related references have been added.

Reviewer 3 Report

Comments and Suggestions for Authors

This manuscript presents a well-designed field study investigating the effects of nitrogen fertilizer rates on spring maize yield and soil nitrogen balance under straw return conditions in the cold black soil region of Northeast China. Here are some comments for improvement.

[1] Clearly state the nitrogen content of the straw or the total amount returned per hectare in Section 2.2 or 2.4.

[2] Provide a reference or justification for the use of 52.65% and 70.66% release rates for straw nitrogen in the first and second years.

[3] Discuss potential reasons of the differences between 2023 and 2024 results in the Discussion.

[4] Why is a 20% nitrogen surplus referenced as an environmental threshold?

[5] A thorough proofread is needed.

[6] Figures 1, and 4-9 are referenced in the text but not included in the provided content.

[7] Double check the tables, equations and references.

Author Response

Dear Reviewer,

Thank you very much for your valuable comments on the paper. We have made modifications to the paper based on your comments, and the parts of the article that have been modified are marked in red font.Please review them. Thanks again!

 Description of modification are as follows

3.Open Review

3.1 Question: Clearly state the nitrogen content of the straw or the total amount returned per hectare in Section 2.2 or 2.4.

Answer: Thank you for ponting this out. We agree with this comment. We have clearly stated the nitrogen content of the straw or the total amount returned per hectare in Section 2.2 and 2.4. (Modification was noted by red in 2.Materials and Methods).

3.2 Question: Provide a reference or justification for the use of 52.65% and 70.66% release rates for straw nitrogen in the first and second years.

Answer: A reference or justification for the use of 52.65% and 70.66% release rates for straw nitrogen in the first and second years has provided in 2.2 Experimental Design

3.3 Question: Discuss potential reasons of the differences between 2023 and 2024 results in the Discussion.

Answer: Potential reasons of the differences between 2023 and 2024 have been supplemented in discustion.(lines 473-477)

3.4 Question: Why is a 20% nitrogen surplus referenced as an environmental threshold?

Answer: Some studies(Li,R.G,2003;Di,H.J,2008;Xia,X.J,2011) suggested that when the nitrogen balance surplus in farmland exceeds 20.0%, it may pose a potential threat to the environment. So a 20% nitrogen surplus was referenced as an environmental threshold in this research. (lines 428-430) 

3.5 Question: A thorough proofread is needed. Figures 1, and 4-9 are referenced in the text but not included in the provided content.Double check the tables, equations and references.

Answer: We have modified in the article. We have check the tables, equations and references.

Round 2

Reviewer 2 Report

Comments and Suggestions for Authors

Authors have carried out a comprehensive review and have responded to most of my comments. However, two issues need attention.

(i) The abstract is lengthy. Authors should reduce the words from the present 495, to at most 300, preferably 250.

(ii) It is not clear whether the experiment was replicated and if so, how many replications. 

Author Response

Dear Reviewer,

Thank you very much for your valuable comments on the paper. We have made modifications to the paper based on your comments, and the parts of the article that have been modified are marked in red font.Please review them. Thanks again!

 Description of modification are as follows

Question 1: The abstract is lengthy. Authors should reduce the words from the present 495, to at most 300, preferably 250.

Answer 1:We have reduced the abstract to less than 300 words.(lines 12-33).

Question 2: It is not clear whether the experiment was replicated and if so, how many replications. 

Answer 2:Three replications for each treatment.(lines 104-105).
